# A Focus on the Proximal Tubule Dysfunction in Dent Disease Type 1

**DOI:** 10.3390/genes15091175

**Published:** 2024-09-07

**Authors:** Elise de Combiens, Imene Bouchra Sakhi, Stéphane Lourdel

**Affiliations:** 1Laboratoire de Physiologie Rénale et Tubulopathies, Centre de Recherche des Cordeliers, Institut National de la Santé et de la Recherche Médicale (INSERM), Sorbonne Université, Université Paris Cité, F-75006 Paris, France; elise.de_combiens@sorbonne-universite.fr (E.d.C.); stephane.lourdel@sorbonne-universite.fr (S.L.); 2Unité Métabolisme et Physiologie Rénale, Centre National de la Recherche Scientifique (CNRS) EMR8228, F-75006 Paris, France; 3Institute of Anatomy, University of Zurich, CH-8057 Zurich, Switzerland

**Keywords:** Dent disease type 1, proximal tubule, metabolism, chronic kidney disease

## Abstract

Dent disease type 1 is a rare X-linked recessive inherited renal disorder affecting mainly young males, generally leading to end-stage renal failure and for which there is no cure. It is caused by inactivating mutations in the gene encoding ClC-5, a 2Cl^−^/H^+^ exchanger found on endosomes in the renal proximal tubule. This transporter participates in reabsorbing all filtered plasma proteins, which justifies why proteinuria is commonly observed when ClC-5 is defective. In the context of Dent disease type 1, a proximal tubule dedifferentiation was shown to be accompanied by a dysfunctional cell metabolism. However, the exact mechanisms linking such alterations to chronic kidney disease are still unclear. In this review, we gather knowledge from several Dent disease type 1 models to summarize the current hypotheses generated to understand the progression of this disorder. We also highlight some urinary biomarkers for Dent disease type 1 suggested in different studies.

## 1. Introduction

The proximal tubule is the most metabolically active segment of the nephron. It reabsorbs 100% of the load of glucose and a significant part of other solutes such as sodium and calcium which are filtered from the blood into the urinary space. This explains why proximal tubule cells contain a high number of mitochondria that provide them with ATP, an energy source that is required for active solute transepithelial reabsorption [1,2]. Another major function of the proximal tubule is the reabsorption of low-molecular-weight proteins via receptor-mediated endocytosis. When dysfunctional, this may lead to growth retardation due to the urinary loss of vitamin-binding proteins [3]. Such abnormality has been observed in patients with Dent disease type 1, a rare X-linked recessive inherited renal disorder affecting mainly males and characterized by the urinary loss of low molecular weight proteins, calcium, and other solutes such as glucose or phosphate, usually associated with a proximal tubule dysfunction. Diagnosed during childhood with the detection of proteinuria, rickets, nephrolithiasis, or due to family history, the disease is progressive and leads to end-stage renal failure, therefore requiring dialysis or kidney transplantation [3,4]. Dent disease type 1 is characterized by inactivating mutations of the *CLCN5* gene encoding the 2Cl^−^/H^+^ exchanger (ClC-5) and is observed in about 65% of patients with Dent disease. Among these patients, 15% present Dent disease type 2 with an *OCRL1* inactivation mutation, and 25% show a typical Dent disease phenotype from unknown origin [3,5]. Over 500 families carrying Dent disease-causing mutations were described; among them almost 200 different mutations of *CLCN5* lead to Dent disease type 1 [3,5,6]. Up to now, there is no specific treatment for Dent disease; therefore, the current therapy is aimed at slowing down the progression of some clinical traits (e.g., nephrocalcinosis) [3,5,7,8].

Numerous *in vitro* and *in vivo* (Table 1) models have been generated to decipher the cellular and molecular mechanisms underlying Dent disease type 1. They include ClC-5 knock-out (KO) [9,10,11,12], knock-in (KI) mice harboring the artificial E211A mutation (inducing a Dent disease phenotype but never described in patients) [13] and the pathogenic N340K mutation (described in patients) [14] (Table 1), knock-down of the ClC-5 homolog in *Drosophila melanogaster* [15], cells carrying ClC-5 mutants derived from DD type 1 patients (ciPTECs) [16,17], *Xenopus laevis* oocytes, and different stably or transiently transfected cell lines [12,18,19,20,21,22,23,24,25].

Although some of the studies performed using these different models of Dent disease type 1 revealed progressive proximal tubule dysfunction with parallel impairment of cell metabolism, the precise role of ClC-5 in these alterations remains yet to be understood. In this review, we highlight past and newly generated hypotheses to explain the pathogenesis of Dent disease type 1—from the onset to the progression to chronic kidney disease—that have emerged from the different experimental models.

## 2. A Dysfunctional ClC-5 Leads to Proximal Tubule Dedifferentiation

Previous studies in the context of Dent disease type 1 made it possible to explore the role of ClC-5 in maintaining an optimal endocytosis of proteins in proximal tubule cells. Indeed, when this transporter is dysfunctional, proteinuria appears as a consequence of reduced protein uptake in this segment of the nephron [4,9,10,31]. Due to its colocalization with the vacuolar H^+^ ATPase, described as the endo-lysosomal acidifying pump, it has been for a long time suspected of creating an electrical shunt for the pump to keep working [31,32]. Hence, the accumulation of Cl^−^ in early endosomes may potentiate endosomal acidification that seems necessary for several processes, from receptor–ligand dissociation to creating an optimal environment for protein digestion in lysosomes [31,33,34,35]. Over the years, ClC-5 physical interaction with cytoskeleton proteins playing a crucial role in endocytosis mechanisms was also highlighted, suggesting that this transporter may participate in endocytosis independently from its electrical activity [36,37]. However, some pathogenic mutants of ClC-5 were described as correctly expressed but bearing impaired electrical activity [20,21,22,23,38]; while patients carrying these mutants still presented proteinuria, the altered subcellular localization of their partners was never validated. Among these mutants, the artificial mutation of ClC-5 E211A converting the transporter in a pure Cl^−^ conductance was induced in a mouse model [13,31]. These mice showed a typical Dent disease phenotype with impaired receptor-mediated endocytosis but normal endosomal acidification, revealing, for the first time *in vivo*, that endocytosis can be inefficient even with normal endo-lysosomal pH evolution [13,22,31,39] (Table 1). More recently, endosomal pH and Cl^−^ concentration were measured in a highly differentiated proximal tubule cell line expressing a similar ClC-5 mutant and showed an impairment of both parameters [12]. Therefore, ClC-5’s exact contribution to maintaining an efficient endocytosis is still an unresolved matter and a combination of all previously mentioned roles of this transporter could be important for optimal protein uptake.

As previous studies aiming at characterizing the real impact of ClC-5 dysfunction on receptor-mediated endocytosis relied solely on end-time endocytosis assays [9,10,13,14], the Hall team recently explored this phenomenon in ClC-5 KO mice using intravital microscopy [30]. These experiments revealed that the endocytosis of filtered proteins is not abolished in these animals but rather slowed down as endo-lysosome trafficking seems to be impaired [30] (Table 1). Moreover, past data from Dent disease patients and ClC-5 KO animals further validate that the endo-lysosomal apparatus does not seem to be altered and multi-ligand receptors are not down-regulated in proximal tubules but instead less recycled back at the plasma membrane [12,40,41] (Table 1). In the recent pathogenic N340K mutant model, protein endocytosis was shown to be extended to later proximal tubule segments, indicating a nephron plasticity in trying to optimize cell function in disease states [14]. Moreover, lysosomes—the final compartments of the endo-lysosomal system but also essential organelles for optimal autophagy within the cell—displayed overall altered distribution in the proximal tubules [14] (Figure 1). Lysosomal hydrolases responsible for protein degradation such as cathepsins were also shown to be abnormally distributed and expressed in mouse models [14,42]. Many cathepsins need further processing when traveling along the endo-lysosomal pathway and a final activating cleavage occurs in optimally acidified lysosomes [43,44,45,46].

In addition to these observations, other major transporters necessary for the reabsorption of electrolytes and metabolites were shown to be less available at the apical plasma membrane; hence, justifying the loss of molecules typically observed when proximal tubule dysfunction occurs [3,9,13,14]. An epithelial-to-mesenchymal transition was also observed from transcriptomic data and tissue immunostainings supporting even more the proximal tubule dedifferentiation taking place in the context of Dent disease type 1 [14] (Table 1). As a consequence of this phenomenon, proximal tubule cell apoptosis and replacement proliferation were observed and accompanied by fibrosis and inflammation, ultimately leading to chronic renal failure in aging animals and some patients [14,29,47,48]. In mice presenting a clear proximal tubule dedifferentiation and proliferation, oxidative stress was observed, which suggests that proximal tubule metabolism is also impacted during Dent disease type 1 [14,29] (Table 1). Interestingly, even in models with a mutant ClC-5 retained within the endoplasmic reticulum, no unfolded protein response could be detected, suggesting that this other form of cell stress does not contribute to the observed oxidative stress [14,17].

## 3. An Alteration of Proximal Tubule Cell Metabolism Is Observed in the Context of Dent Disease Type 1

Cell mitochondrial abundance depends on its energy needs and, as previously mentioned, the proximal tubule requires enough energy production to maintain its massive absorptive functions [49]. In addition to ATP production, mitochondria maintain intracellular calcium concentrations, redox homeostasis, and participate in regulating cell apoptosis [50,51,52]. Thus, in several pathological contexts affecting the proximal tubule, mitochondrial dysfunctions have been observed [53,54,55]. Regarding Dent disease type 1, the pathogenic N340K mouse model shows, for the first time, clear signs of mitochondrial metabolism alterations [14]. Indeed, a decreased expression of genes involved in oxidative phosphorylation and fatty acid metabolism as well as a significant urinary loss and altered renal cortex handling of mitochondrial metabolites were highlighted in this model [14] (Table 1). We are currently exploring whether similar phenomena are occurring in Dent patients by analyzing their urine content. Furthermore, the *Clcn5* KO model studied by Wright et al. displayed altered gene expression related to proximal tubule metabolism and carbohydrate homeostasis, which could impact gluconeogenesis [27,56] (Table 1). Thus, other metabolic pathways should be further explored. Indeed, a possible way for cells to show changes in the functioning of their ATP-producing Krebs cycle could be by altering gluconeogenesis, which is responsible for regulating the cataplerosis process [56,57]. Apart from the common glucosuria observed in Dent disease patients and animal models, no alteration in glycemia was communicated, but this phenomenon could be adjusted at the systemic level. Indeed, the pathogenic N340K mouse model showed a clear reduction in their fat mass, suggesting a decreased energy storage or an increased need for energy release [14].

Mitochondrial dysfunction may compromise the amounts of ATP generated and contribute to increased oxidative stress via the mitochondrial respiratory chain [58]. This would, in turn, cause damage to cellular components such as proteins, lipids, and DNA, promote mitochondrial fragmentation, and activate pro-inflammatory pathways [59]. In cells and animal models for Dent disease type 1, increased antioxidant defenses were observed, with an upregulation of SOD1 and Thioredoxin-encoding genes, activation of ROS metabolic processes, and in some cases even a mild increase in protein carboxylation [14,17,27,28,29] (Table 1). The role of albumin in controlling oxidative stress was also suggested. Indeed, this protein could show a protective role in proximal tubules by binding to ROS. In Dent disease type 1, a decreased endocytosis of albumin could compromise this function and participate in the observed oxidative stress [60,61].

In addition to the potentially altered functioning of proximal tubule mitochondria, lipid metabolism was also shown to be impacted in the context of Dent disease type 1 [14,27]. Fatty acid β-oxidation is the main pathway used to produce energy in proximal tubule mitochondria. When β-oxidation is reduced and the cell’s capacity to oxidize fatty acids is exceeded, these molecules are stored in the form of lipid droplets, and can—in excess—cause lipotoxicity and eventually have an impact on renal function [62]. The accumulation of these lipid droplets has been observed in models developing chronic kidney disease, including in the pathogenic N340K mouse model for Dent disease type 1, and is thought to participate in ROS production, thus promoting fibrosis and inflammation, ultimately leading to chronic renal failure [14,27,62,63] (Figure 1). On the other hand, studies in cultured cell models for Dent disease type 1 have not highlighted any alteration in lipid metabolism, suggesting that these mechanisms occur over a longer time in integrated models [17].

Furthermore, it was shown that plasma membrane lipid and cholesterol composition can modify not only the electrical activity of transporters but also the dynamics of vesicle trafficking [4,64,65]. Hence, as already suggested by colleagues, the altered lipid metabolism observed in animal models for Dent disease type 1 could play a role in reduced endosomal recycling and trafficking [4,14,27]. Up to now, it has been difficult to pinpoint a cause for this altered lipid metabolism other than an initial mitochondrial impairment. The latter could be a consequence of a defect in clearing damaged mitochondria due to the endo-lysosomal system inefficiency (altered mitophagy). It could also be caused by a reduced uptake of metabolites necessary for cell metabolism following the decreased apical availability of their transporters (impaired recycling) [14] (Table 1). Another possibility could be a direct interaction between the endo-lysosomal system and lipid droplets—that would be impaired in the context of Dent disease type 1—as such a phenomenon was already observed in hepatocytes using high-resolution microscopy [66].

Although the exact mechanisms behind the altered proximal tubule cell metabolism are yet to be unraveled, previous transcriptomic, proteomic, and metabolomic studies have made it possible to shed light on potential biomarkers for Dent disease type 1 and its evolution to more severe forms.

## 4. Studies Revealed a Diversity of Potential Biomarkers for the Progression of Dent Disease Type 1

In 2008, the carbonic anhydrase type III (CA III) kidney-specific upregulation was described in the context of Dent disease type 1 (patient and *Clcn5* KO mice) [29] (Table 1). This enzyme catalyzes the production of HCO_3_^−^ and H^+^ in skeletal muscles, adipose tissue, and, to a lesser extent, in the liver and the kidney [29,67,68]. While it does not seem to play a crucial role in pH regulation, it was shown to be induced and remains stable under oxidative stress conditions, where it acts as an ROS scavenger, hence protecting cells against the bad sides of such stress [68,69]. Its encoding gene *Car3* was shown to be overexpressed in Dent disease type 1 models inducing a secretion of CA III from dedifferentiating proximal tubules [14,27,29] (Table 1). Moreover, this protein was found in slightly increased levels in urines from megalin KO mice, presenting altered endo-lysosomal function and dedifferentiating tubules as well [29]. Hence, this urinary marker could be used as a reflector of the oxidative stress level within proximal tubules and their consequent dysfunction, even if it does not seem to be specific to Dent disease [29].

α-ketoglutarate (αKG) is responsible for the regulation of the Krebs cycle and its substrates. It participates in amino acid synthesis, ATP production, the respiratory chain functioning, and can modulate oxidative stress [70,71]. It also plays a role in the hydroxylation of proteins, nucleic acids, lipids, and metabolites, as well as cell proliferation and differentiation through calcium-mediated mechanisms [72,73]. In young pathogenic N340K mice, cortical and urinary αKG levels were increased, which could play a significant role in protecting the nephron at this disease stage, whereas with time, these effects would not be sufficient to contain the tissular damages observed [14]. The excretion of this metabolite could either reflect the activation of defense mechanisms or the consequence of an uncontrolled proximal tubule dysfunction. Further studies would be needed to evaluate the potential of using αKG as a urinary biomarker of proximal tubule metabolic alteration in Dent disease type 1.

In the kidney cortex, the hormone serotonin, which is synthesized from the amino acid tryptophan, participates in the regulation of glomerular filtration, vascular resistance, and mitochondrial biogenesis to improve renal function [74,75,76]. However, it was also suggested that high levels of serotonin promote proximal tubule epithelial-to-mesenchymal transition, local inflammation, and tissue damage [75]. In a current analysis of urines from patients with Dent disease, serotonin was generally found in very low amounts (unpublished data). Moreover, young pathogenic N340K mice, not displaying chronic kidney disease, presented reduced serotonin in their urine paralleled by increased urinary tryptophan [14]. As Dent disease type 1 is characterized by the altered uptake of amino acids, this could justify the reduced availability of tryptophan to produce serotonin. Hence, this marker does not seem to be playing a role in the proximal tubule dedifferentiation and fibrosis observed. Other metabolites could be more implicated in these phenomena in the context of Dent disease type 1.

*In vivo*, inosine is generated from the less stable adenosine nucleoside and was shown to significantly increase in body fluids in several pathological contexts [77,78,79,80]. Both molecules can bind to the same receptor A_2_A-R but would activate different pathways; inosine would stimulate ERK1/2-mediated proliferation and apoptosis of cells whereas adenosine would activate a cAMP pathway with opposite consequences [81,82,83]. The inosine to adenosine ratio could therefore be an important parameter to modulate the immune response and associated tissue damage [81]. In the urines from the pathogenic N340K mouse model, this ratio was increased, which could participate in the tubular cell apoptosis, progressive inflammation, and fibrosis observed [14]. Further studies are required to explore this urinary ratio in patients with Dent disorder at various disease states.

In different Dent disease type 1 models, NGAL/Lipocalin 2 was shown to be overexpressed and even excreted [14,26] (Table 1). In past studies, it was hypothesized that this protein is overexpressed by distal tubules as a result of the increased presence of retinoic acid there. Indeed, the latter would be less taken up by proximal tubules due to the reduced endocytosis of its binding protein RBP. Reaching the more distal tubules, it would freely diffuse through the plasma membrane to stimulate the expression of its target genes, including *Lcn2* [26,84]. More recently, Sakhi et al. showed a proximal tubule localization of NGAL in young animals that was switched to the distal nephron in older N340K mice presenting chronic kidney disease. Hence, the authors hypothesized that proximal tubules express this protein due to the increased local oxidative stress and that the circulating NGAL activates inflammation and fibrosis pathways through its distal receptor 24p3R [14]. These increased urinary NGAL levels were observed in different renal diseases and are thought to directly reflect renal function [85]. This biomarker could then be used in the Dent disease type 1 context to evaluate the level of renal damage patients are experiencing.

## 5. Current State of the Art—Hypotheses on Dent Disease Type 1 Progression

Experimental models designed for Dent disease type 1 study made it possible to pinpoint several potential pathogenetic mechanisms involved in this disorder [9,10,11,12,13,14,15,16,17,18,19,20,21,22,23,24,25,48]. Some observations are clearly defined: in the proximal tubule, ClC-5 is necessary for the proper functioning of receptor-mediated endocytosis of plasma-filtered proteins, and when this phenomenon is altered over a long time, proximal tubules dedifferentiate, and progressive tissue damage occurs (Figure 1) [3,9,10,12,13,18,27,29,31,48,61]. However, the exact role of ClC-5 in protein uptake remains unclear. As colleagues showed that endocytosis could be disrupted even with normal pH gradients, they hypothesized that an optimal Cl^−^ accumulation in endosomes is essential for endosomal Ca^2+^ transport that would itself intervene in vesicle trafficking and functioning [13,31,86,87]. New emerging tools coupled with high-resolution microscopy could help answer this question and define whether ClC-5’s contribution to efficient endocytosis lies in its electrical activity and/or its physical interaction with crucial partners.

In several models, renal oxidative stress and metabolic disorders have been observed [14,26,27,29]. In 2015, based on observations made in other contexts, colleagues hypothesized that it could be a consequence of altered autophagy and dedifferentiating proximal tubules [61]. Indeed, excessive ROS production from damaged and non-cleared mitochondria could disrupt cell junction and consequently have an impact on epithelium integrity [88]. A free zona ocludens 1 would then release its binding protein, the transcription factor ZONAB, that would upregulate (e.g., cyclin D1) or downregulate (e.g., megalin) its target genes [89,90]. This would cause cell proliferation and dedifferentiation, as observed in some models for Dent disease type 1 [14,29,61]. In the latest mouse model for this disorder, no alteration of cell junctions was observed, and the multi-ligand receptor was not downregulated at the gene level, but rather less recycled back at the plasma membrane and probably rerouted to degradative pathways [14]. In addition, no clear accumulation of damaged mitochondria has been shown so far, and the autophagy defect was also not directly defined. Even though the distribution of lysosomes is abnormal, and some markers of autophagy suggest this pathway is impacted [14], no exploration of this mechanism in dynamic conditions has been conducted yet (Figure 1). Up to now, we only know that metabolites that are crucial for mitochondrial function are more excreted in the context of Dent disease type 1 [14] (and unpublished data); meanwhile, genes encoding proteins contributing to cell metabolism seem to be downregulated [14,27]. Moreover, lipid metabolism also appears to be impacted, showing fatty acids that are less excreted in urines from Dent disease type 1 mice and lipid droplets that accumulate (Figure 1) [14,27]. In other contexts, these parameters usually reflect mitochondria dysfunction [62,91].

As highlighted in the biomarker section, several metabolites and proteins are found in increased amounts in kidneys and urines from mice affected with Dent disease type 1. Many of them could reflect the cell oxidative stress status and the progression of fibrosis and inflammation in the kidneys of animals that start to show signs of renal failure [14]. Among them, NGAL/Lipocalin-2 could even play pro-inflammatory and pro-fibrotic roles itself as it could trigger its distal receptor 24p3R while being in excess in the nephron lumen (Figure 1). NGAL was also shown to stimulate the production of collagens and profibrotic molecules *ex vivo* [92]. However, this should be explored further in animals and patient urines. In addition, using a cultured proximal tubule cell model, colleagues recently demonstrated that some collagens can be overexpressed, and their lysosomal degradation impaired in the context of Dent disease type 1; hence, justifying an increased fibrosis consequent to the altered endo-lysosomal function [48]. In closing, the interstitial fibrosis observed in patients and animal models for Dent disease type 1 could have multiple origins. It could be triggered by dedifferentiating proximal tubules, potentiated by a local oxidative stress, and/or even maintained by metabolites activating distal pathways.

## 6. Conclusions

In conclusion, even though a wide range of cell and animal models were designed to study Dent disease type 1, some uncertainties regarding its pathogenesis remain. As highlighted in Table 1, animal models for this disorder show variability in their phenotype and proximal tubule dysfunction. Moreover, past clinical studies could not draw any correlation between genotype and phenotype as patients carrying the same mutation would present with different clinical traits [3,23]. It was suggested that environmental factors could heavily contribute to this variability, rendering the clinical applications of findings from laboratory models even more challenging [26]. Although understanding the exact molecular mechanisms underlying Dent disease type 1 pathogenesis needs further exploration, the emergence of non-invasive biomarkers shows promise as they could serve as predictors of the actual severity of one patient’s phenotype. Indeed, as shown in CKD contexts, urinary NGAL can efficiently reflect a patient’s GFR and it starts to be over- excreted even in N340K animals not presenting CKD yet [14,93,94,95,96]. Hence, assessing this marker could allow us to estimate how close one is to developing end-stage renal disease. Therefore, involving the exploration of patient urines on a larger cohort would validate the relevance of these biomarkers and perhaps someday help to predict patient’s renal outcome.

As several new techniques are emerging within the field, some molecular mechanisms of Dent disease type 1 pathogenesis could become clearer in the following years. Therefore, ways of slowing disease progression could emanate from this research and find a clinical application. As suggested by recent data, a mitochondrial dysfunction seems to occur over the course of this disorder [14], and this mechanism was described as a key player in the development of various forms of CKD [97,98]. Several kidney diseases involving mitochondrial dysfunction showed a slowed down progression when attenuating the consequences of impaired mitochondria using mitochondria-targeted therapeutic approaches [99]. Some of them—such as mito-TEMPO, mitoQ, or other antioxidants—have already shown promising results in bringing down the occurrence of kidney tissue damage, even in the case of a mitochondrial dysfunction caused by impaired lysosomes [54,100,101,102,103,104]. On the other hand, the stimulation of autophagy was also described as a potential therapeutic approach for some neurodegenerative diseases as it might reduce the observed oxidative stress [105]. Hence, testing these possible treatment options in laboratory models would be an interesting first step in trying to slow down Dent disease type 1 progression.

Moreover, strategies aimed at correcting ClC-5 dysfunction in Dent disease might be another potential promising therapeutic strategy. For example, Yadav et al. targeted by lentiviral strategy the *CLCN5* cDNA into the kidney of ClC-5 KO mice and observed increased megalin expression, improved diuresis, and decreased urinary calcium and protein excretion during a few months. Thus, gene therapy was effective in ameliorating some traits of Dent disease type 1 but body tolerance needs improving for the effects to last over a longer time [11]. Gabriel et al. also showed an improved proximal tubule function, with decreased low-molecular-weight proteinuria, glycosuria, calciuria, and polyuria for several months after transplantation of WT bone marrow in ClC-5 KO mice [106].

Furthermore, it is well known that proteins that do not reach their functional destination can be rescued by pharmacological approaches. For instance, the combination of ivacaftor, tezacaftor, and elexacaftorin strongly increases currents and plasma membrane expression of the most frequent mutation of CFTR, causing cystic fibrosis due to abnormal processing of the mutant protein (ΔF508) [107]. This strategy might be adapted to Dent disease type 1, when the mutant ClC-5 protein is retained in the endoplasmic reticulum, and therefore not able to reach the early endosomes and the plasma membrane [23,38].

Overall, several therapeutic approaches could be envisaged to correct Dent disease phenotype in mice—from the restoration of a functional ClC-5 to preventing the tissue damage to occur and the disease to progress—but understanding the phenotypical variability should remain a research focus for future clinical applications.

## Figures and Tables

**Figure 1 genes-15-01175-f001:**
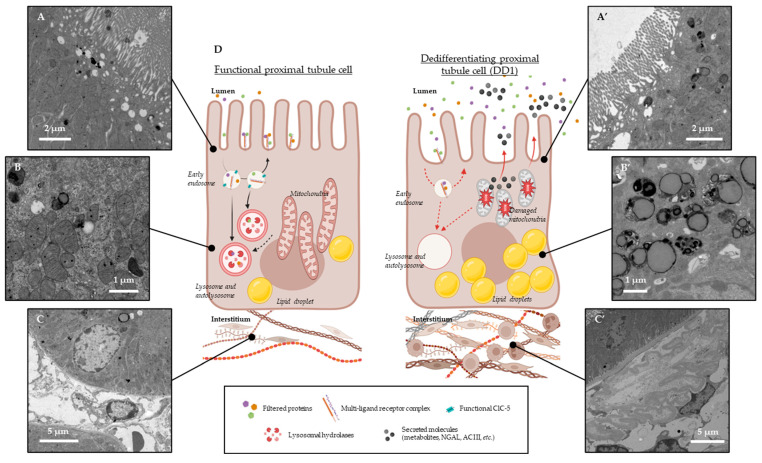
Histological signs of proximal tubule dedifferentiation in the context of Dent disease type 1 (**A**–**C**, **A’**–**C’**) and hypothesis on the pathogenesis of Dent disease type 1 (DD1) (**D**). Electron microscopy images from 10-month-old WT (**left**, **A**–**C**) and N340K mice (**right**, **A’**–**C’**) reflecting (**A’**) a reduced availability of endosomes and lysosomes, (**B’**) an accumulation of lipid droplets, and (**C’**) an increased interstitial fibrosis in the context of Dent disease type 1 in comparison to physiological states (**A**–**C**). (**D**) In physiological contexts (left diagram), the proximal tubule reabsorbs filtered plasma proteins using receptor-mediated endocytosis (through a multi-ligand receptor complex composed of megalin, cubilin, and ammionless). Once attached to their receptors, proteins are internalized within the cell and reach a first endosomal compartment where a drop in pH occurs thanks to a functional ClC-5 transporter. In these early endosomes, proteins detach from their receptors and further traffic along the endo-lysosomal apparatus until reaching lysosomes, while receptors are recycled back to the plasma membrane to catch other filtered proteins. The highly acidified lysosomes enriched with functional hydrolases called cathepsins take care of digesting these proteins. According to Polesel et al., the degradation product is secreted back to the lumen, and the resulting peptides are further processed by later proximal tubule segments [30]. On the other hand, functional lysosomes also fuse with autophagosomes to ensure damaged organelle clearance for general cell health. In the context of Dent disease type 1 (right diagram), the absence of a functional ClC-5 on the early endosome membrane would prevent the early acidification step from occurring; hence, uptaken proteins would hardly detach from their receptors which would in turn slow down the internalization of additional proteins, the recycling of multi-ligand receptors, and the further trafficking of vesicles along the endo-lysosomal pathway. Additionally, lysosomes would lack functional cathepsins as these proteases need an optimal endo-lysosomal system to be efficient. Transporters that would be internalized on early endosomes would also be less recycled back at the plasma membrane, which would reduce the reabsorption of electrolytes and other molecules in proximal tubules. Stressed mitochondria lacking their essential metabolites to ensure their functions would accumulate within the cells but without being effectively cleared by autophagy. Lipids would also accumulate within the proximal tubules as they would be harder to process by the dysfunctional cells. The consecutive increased oxidative stress would activate stress pathways and the production of damage-associated biomarkers. A renal function decline would be observed as the tissue’s local inflammation and fibrosis progress (with an activation of fibroblasts in the environing interstitium and the infiltration of immune cells). The dysfunctional proximal tubule cells would slowly activate epithelial-to-mesenchymal transition factors and dedifferentiate.

**Table 1 genes-15-01175-t001:** Summary of the main findings related to proximal tubule dysfunction in animal models for Dent disease type 1.

Animal Model	Study	Main Findings Related to Proximal Tubule Dysfunction
*Clcn5* KO mouse model Jentsch lab	Piwon N et al., 2000 [9] and Novarino G et al., 2010 [13]	- Altered receptor-mediated and fluid phase endocytosis - Reduced apical expression of PT transporters - Altered endosomal acidification
	Maritzen T et al., 2006 [26]	- Altered Vitamin D metabolism and overexpression of several genes in the distal nephron - Overexpression of the genes encoding NGAL and collagen
*Clcn5* KO mouse model Guggino lab	Wang SS et al., 2000 [10]	- Altered receptor-mediated and fluid phase endocytosis
	Wright J et al., 2008 [27] and Guggino SE et al., 2009 [28]	- Changes in gene involved in lipid and cholesterol metabolism, and in carbohydrate homeostasis - Overexpression of *Car3* and alteration in the expression of oxidoreductase activity-related genes
	Gailly P et al., 2008 [29]	- Increased expression of CA III and urinary excretion of CA III - Increased proliferation of kidney cells - Increased expression of oxidative stress markers
	Polesel M et al., 2022 [30]	- Altered endosomal trafficking
*Clcn5^unc^* E211A mouse model	Novarino G et al., 2010 [13]	- Altered receptor mediated and fluid phase endocytosis - Reduced apical expression of PT transporters - Maintained endosomal acidification
*Clcn5* KO mouse model Lu lab	Yadav MK et al., 2022 [11]	- Reduced megalin expression
*Clcn5* KO mouse model Weisz lab	Shipman KE et al., 2023 [12]	- Delay in endosome maturation and reduced ligand degradation
*Clcn5* N340K pathogenic mouse model	Sakhi IB et al., 2024 [14]	- Altered receptor-mediated and fluid phase endocytosis - Dedifferentiation of PT cells and increased renal cell apoptosis - Downregulation of genes involved in oxidative phosphorylation and lipid metabolism - Increased excretion of metabolites related to mitochondrial functions - Overexpression of *Car3* and increased protein carbonylation - Overexpression of *Lcn2* and increased excretion of NGAL

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
