# Peer review of "A Focus on the Proximal Tubule Dysfunction in Dent Disease Type 1"

_genes, 2024, doi:10.3390/genes15091175_

Round 1

Reviewer 1 Report

Comments and Suggestions for Authors

In their review titled “A focus on the metabolic dysfunction of proximal tubule cells in Dent disease type 1” de Combiens et al. have explored and summarized the main findings of studies on different models regarding the molecular mechanisms and pathogenesis of Dent disease type 1, with a focus on changes in the metabolism of proximal tubule cells. They have also reviewed potential urinary biomarkers for Dent disease type 1. The review is well-written; however, I have some suggestions for improving the manuscript.

General concept comments:

1) The introduction section should be slightly expanded with more information about Dent disease. A few statements about the epidemiology (male predominance is mentioned in the abstract but not the main text; is the prevalence/incidence different in different populations worldwide etc.) and the differential diagnosis of Dent disease would give the reader without prior knowledge of Dent disease a better overview. Additionally, while this review is focused on type 1 disease (as mentioned 65% of all cases), a statement regarding type 2 disease (both the mutated gene and percentage of cases) and the percentage of cases without a known cause would also be beneficial.

2) Since the focus of the review is on the metabolic dysfunction of proximal tubule cells in Dent disease type 1, a table containing the studies on the changes of metabolism in Dent disease type 1 would allow for easier access to the main information and the text can be a source of more detailed explanations. The table should contain the studies, models used in the studies and their main findings. An example:

Study

Models used

Main findings

Sakhi IB et al. [13]

N340K mouse model

Decreased expression of oxidative phosphorylation and fatty acid metabolism genes; urinary loss of mitochondrial metabolites

3) Most relevant recent publications were referenced and discussed in this study. I suggest adding a recently published study (April 2024) by Duran et al. “Renal antiporter ClC-5 regulates collagen I/IV through the β-catenin pathway and lysosomal degradation” (DOI: 10.26508/lsa.202302444) as it gives insight into a new mechanism for progression of Dent disease type 1 into renal fibrosis.

Specific comments:

4) Lines 24-26 “It reabsorbs 100% of the filtered load of glucose and a significant part of other solutes such as sodium and calcium which originate from the Bowman’s space.” The solutes don’t originate from Bowman’s space, they are just filtered into it. A more precise phrasing would be “It reabsorbs 100% of the load of glucose and a significant part of other solutes such as sodium and calcium which are filtered from the blood into the urinary space.”

5) Line 31 “vitamin-bound proteins” should be “vitamin-binding proteins”.

6) Line 11 “inherited” and line 32 “hereditary” should be changed to (or just added) “X-linked recessive” as the mode of inheritance is known.

7) Abbreviations – line 37-38 “2Cl-/H+ exchanger” should be followed by “(ClC-5)” to introduce the abbreviation after the full form is first mentioned. Line 128 “TCA cycle” the full form of TCA should be written the first time it is mentioned.

8) Line 43 “non-pathogenic E211G mutation” How is it a non-pathogenic mutation when later in the manuscript it states that E221G mice showed a typical Dent disease phenotype with impaired receptor-mediated endocytosis (Lines 69-73)?

9) Lines 92-95 “Lysosomal hydrolases responsible for protein degradation were also shown to be abnormally distributed and expressed in mouse models [13,37]. Many cathepsins need further processing when traveling along the endo-lysosomal pathway and a final activating cleavage occurs in optimally acidified lysosomes [38–41].” I suggest adding “, such as cathepsins,” after “protein degradation” to better connect the two sentences.

10) Line 109 “suggesting that this other cell stress does not contribute to the observed oxidative stress” I suggest adding “form of” after “other” to make the statement easier to interpret.

11) Line 188 “A-ketoglutarate” should be “α-ketoglutarate”.

12) I suggest adding histological images, if possible, to Figure 1 of both normal proximal tubule cells and proximal tubule cells with visible signs of dedifferentiation as shown in the schematic. Either basic light microscopy, or even better transmission electron microscopy, would allow for better visualization of the changes. If the authors do not personally have microscopic images of the proximal tubule cells, copyright-free images can be obtained and used from other studies, provided they are properly referenced of course, such as the study found at https://doi.org/10.3390/ijms25020966.

Comments on the Quality of English Language

Minor changes to phrasing should be made during the editing process.

Reviewer 2 Report

Comments and Suggestions for Authors

Combiens et al in the review article "A focus on the metabolic dysfunction of proximal tubule cells in Dent disease type1" presented the basics of the physiological changes observed in Dent's disease. Unfortunately, the review article is very superficial and does not address the "metabolic dysfunction" adequately. The review article heavily centers around the lack of protein reabsorption and the endocytic pathways, reactive oxygen species, and autophagy, but not much on the metabolism in proximal tubule cells. Therefore, the title is misleading.

Overall, the manuscript is well written. However, there are one or two sentences that need attention.

Line 68: "we can imagine ......" please clarify the sentence.

Line 113: "energy needs" will read better than "energetic needs"

Line 176-187: Please clarify if the changes in CA-III are specific to Dent's disease or this is observed in more than one conditions where PT function is compromised.

Line 196: remove "be" in the line before "the consequences"

Line 234: The word "Anyways" would be very casual and dismissive.

Reviewer 3 Report

Comments and Suggestions for Authors

This review article discusses the metabolic dysfunction of proximal tubule cells in Dent disease type 1, a rare genetic renal disorder caused by inactivating mutations in the CLCN5 gene. The authors gather insights from various experimental models to provide a comprehensive overview of the mechanisms underlying this disorder, focusing on cellular dedifferentiation, mitochondrial dysfunction, oxidative stress, and lipid metabolism alterations.

The introduction provides an extensive background on Dent disease type 1 and the role of ClC-5 in proximal tubule cells, establishing a clear context for the focus on metabolic dysfunction. The authors give an effective summary of the current state of knowledge on Dent disease type 1, linking it to the wider context of renal dysfunction and chronic kidney disease .

The review method is well-documented, drawing on a wide range of in vitro and in vivo models, including ClC-5 knockout models, as well as data from various bioinformatic and omics studies.

The authors provide a detailed examination of metabolic dysfunctions in proximal tubule cells, focusing on mitochondrial alterations, lipid metabolism, and oxidative stress in Dent disease type 1.The discussion on potential biomarkers, including NGAL/Lipocalin 2, carbonic anhydrase III, and α-ketoglutarate, is insightful and clinically relevant, providing avenues for future diagnostic tools.
The authors do a good job of highlighting areas where knowledge gaps remain, such as the precise contribution of ClC-5 to receptor-mediated endocytosis and the complex interplay between metabolic dysfunctions and proximal tubule dedifferentiation.

This manuscript offers a comprehensive and detailed review of the metabolic dysfunctions associated with Dent disease type 1, particularly focusing on mitochondrial alterations, oxidative stress, and lipid metabolism. The authors have synthesized a broad array of experimental data, providing valuable insights into the pathogenesis of Dent disease type 1. The review is well-structured, methodologically sound, and contributes significantly to the understanding of this rare disorder.

To be frank, I liked the content of the manuscript and the structure, but it missed critical components such as discussion and conclusions, and, as such, resembles more the introduction of a thesis, instead of a review. Include a discussion and a conclusions section. For example, while the manuscript touches on the limitations of the current models, it could be strengthened by a more in-depth analysis of the challenges in translating these findings to clinical applications and patient treatment. The conclusion section should provide more specific recommendations for future research directions, particularly regarding therapeutic strategies to target mitochondrial and metabolic dysfunctions in Dent disease type 1.

Round 2

Reviewer 2 Report

Comments and Suggestions for Authors

Authors have answered all comments.